# Rapid Approach to Determine Propionic and Sorbic Acid Contents in Bread and Bakery Products Using ^1^H NMR Spectroscopy

**DOI:** 10.3390/foods10030526

**Published:** 2021-03-03

**Authors:** Marwa Scharinger, Marcel Kuntz, Andreas Scharinger, Jan Teipel, Thomas Kuballa, Stephan G. Walch, Dirk W. Lachenmeier

**Affiliations:** 1Unit of Research of Plant Ecology, Faculty of Sciences, Campus Academia, University of Tunis El-Manar II, Tunis 2092, Tunisia; marwa.scharinger@outlook.com; 2Chemisches und Veterinäruntersuchungsamt (CVUA) Karlsruhe, D-76187 Karlsruhe, Germany; marcel.kuntz@cvuaka.bwl.de (M.K.); andreas.scharinger@cvuaka.bwl.de (A.S.); jan.teipel@cvuaka.bwl.de (J.T.); thomas.kuballa@cvuaka.bwl.de (T.K.); stephan.walch@cvuaka.bwl.de (S.G.W.)

**Keywords:** bread, NMR spectroscopy, propionic acid, quality control, sorbic acid

## Abstract

The food additive sorbic acid is considered as an effective preservative for certain cereal products, and propionic acid is commonly added in bakery wares, e.g., bread and fine bakery wares. The aim of this study was to develop and validate a new nuclear magnetic resonance spectroscopy (^1^H NMR) method for the routine screening and quantification of sorbic and propionic acids in bread and several bakery products for quality control purposes. Results showed that none of the screened samples contained higher concentrations than regulatory maximum limits. However, for some samples, labelling of preservatives was lacking or they were used in food categories, for which the use is not approved. It can be concluded that the developed NMR method can be used for the routine screening of bakery products.

## 1. Introduction

Bread has been considered as one of the most important human staple foods. The use of food additives to enhance shelf life also dates back to ancient history [1]. The World Health Organization (WHO) and the Food and Agriculture Organization (FAO) of the United Nations have defined food additives as substances “which are added intentionally to food, generally in small quantities, to improve its appearance, flavor, texture, or storage properties” [2,3].

It has been postulated that sorbic acid and propionic acid may be safely used as antimicrobial agents (Annex II to EC regulation number 1333/2008) [4,5]. Both food additives may be used in prepackaged sliced bread and rye bread. In addition, its use is also allowed in pre-baked and packaged bakery products, as well as in energy-reduced bread, partially baked, prepacked bread and prepacked rolls, tortilla and pitta, boller and dansk flutes [4].

Sorbic acid (2,4-hexadienoic acid, E 200) is a straight-chain monocarboxylic acid, while potassium sorbate (E 202) is its potassium salt. Sorbic acid and potassium sorbate are white and crystalline powders. Potassium sorbate is soluble in alcohol and freely soluble in water, while sorbic acid is less soluble in alcohol and ether and only slightly soluble in water [6].

The safety of propionic acid (E 280) and its salts (E 281–283) as a food additive was re-evaluated in 2019 by the European Food Safety Authority (EFSA) Panel. No safety concerns for consumers were identified [7].

In the EU, propionic acid has maximum permitted levels for use in bread and rolls ranging from 1000 to 3000 mg/kg [4]. Both the EU Scientific Committee for Food and the Joint FAO/WHO Expert Committee on Food Additives have evaluated propionic acid as food additive on several occasions in 1974, 1973 and in 1990. The latter defined the acceptable daily intake as “not limited” for propionic acid and its sodium, potassium, and calcium salts [5,8]. Nevertheless, EU legislation allows the use only in certain food categories when specified maximum levels are upheld (Table 1).

Nuclear magnetic resonance spectroscopy has been developed for routine quality control as well as the detection of harmful substances in various foodstuffs, such as alcoholic beverages and coffee [9]. The NMR technique provides replicable data and elucidates definite fingerprints that have been proven suitable for authenticity testing. Thus far, only high-performance liquid chromatography, gas chromatography and gas chromatography/mass spectrometry are the validated analytical methods for the determination of preservatives and food contaminants, including cereal products [10,11,12]. However, NMR has several advantages, such as the small amounts required for sample preparation and the reduced time of experimentation. It can be regarded as a key method for the quantification of specific compounds from different food matrixes. All parameters that are fundamental to calculate a certain concentration out of the spectrum are directly provided by the NMR-experiment, the chemical identity of the molecule investigated, and its quantity using an internal or external reference [13,14].

The aim of this study was to extract sorbic and propionic acids from the complex bread matrix in a short time by automated steam distillation (less than five minutes) and to quantify them precisely and reliably using ^1^H NMR for quality control and food surveillance purposes.

## 2. Materials and Methods

### 2.1. Chemicals

Reagents and standard compounds were of analytical or HPLC grade. Deuterated water (D_2_O) was purchased from Deutero GmbH (Kastellaun, Germany). The 3-(trimethylsilyl)propion-2,2,3,3-d_4_ acid, Na salt (TSP), sulfuric acid (H_2_SO_4_), potassium disulfate (KHSO_4_), chemicals to prepare a buffer solution (mixture of 10 mL of D_2_O, 7.5 g of monosodium phosphate (NaH_2_PO_4_), 1000 mg of phosphoric acid (H_3_PO_4_ 85%) with a pH of 3.2) and propionic and sorbic acids standard solutions were all purchased from Sigma-Aldrich (Steinheim, Germany).

### 2.2. Sample Preparation

Bread and bakery products samples were randomly selected from German markets on several occasions for two projects conducted in 2017 and 2019. Samples were stored at −18 °C before analysis. The sample preparation was based on steam distillation specified in the German Official Method for the determination of propionic acid in bread [15]. However, automated instead of manual steam distillation was applied. To achieve this, 10 g of ground sample was filled up into a 750 mL Kjeldahl flask of the automated distillation device (Vapodest 200, Gerhardt GmbH & Co. KG, Königswinter, Germany) set at 75% of steam power. For more details on the automated steam distillation device, see Lachenmeier et al. [16].

Then, a mixture of 10 g of KHSO_4_, 0.5 mL of sulfuric acid (7.1 M) and 3 mL of D_2_O were added. Each distillation into a volumetric flask prefilled with some water to encompass the outlet tube lasted four minutes; following that the volumetric flask was adjusted to 100 mL with demineralized water. After shaking of the volumetric flask, 500 μL of the distillate was pipetted into an NMR tube and mixed thereafter with 60 μL of TSP/D_2_O solution (10 mg/mL) and 60 µL of buffer solution.

### 2.3. NMR Analysis at 400 MHz

All NMR measurements were performed on a Bruker-Avance 400 Ultra-shield spectrometer (Bruker Biospin, Rheinstetten, Germany) equipped with a 5 mm selective inverse probe (SEI) with Z-gradient coils, using a Bruker Automatic Sample Changer (B-ACS 60, Bruker Biospin, Rheinstetten, Germany). The ^1^H NMR spectra were acquired using the noesygppr1d_d7 pulse program with a time domain (TD) of 131072, 4 dummy scans (DS), 32 scans (NS), sweep width of 20.5617 ppm (SW), the spectral width in Hertz was 8223.685 Hz (SWH), additional delay (D7) of 50 s, and an acquisition time of 7.696 s. Receiver gain (RG) was set to 32 and a relaxation delay (D1) of 4 s. A pulse calibration for optimization was performed before every measurement. The pulse calibration program for the optimization of pulse lengths, the size of the processed data (SI) was 262,144, the window multiplication mode was exponential (WDW EM), and the Lorentzian broadening factor for exponential window multiplication (LB) was 0.3 Hz. The spectra were automatically phased and base-line corrected using Topspin version 3.2 and 3.5 (Bruker Biospin, Rheinstetten, Germany).

### 2.4. Validation

Propionic acid and sorbic acid were added to 10 g of bread. The final spiking levels were 250, 500, 1000, 2000 and 3000 mg/kg. The spiked samples were prepared as described above in Section 2.2. For the determination of linearity, limits of detection, quantification, and recoveries, four concentration levels ranging from 200 to 2500 mg/kg were used. Each of them was prepared five times, measured, and evaluated.

### 2.5. Data Analysis and Quality Control

Peak areas in the ^1^H-NMR spectra were evaluated using a MatLab script. The peak areas were set on using trapezoidal integration. Quantification was performed using the Electronic-REference To access In vivo Concentrations (ERETIC) factor, as previously described [9].

## 3. Results

### 3.1. Validation Results

The detection limits of propionic and sorbic acids together with the concentration ranges, and the expected and obtained concentrations expressed in mg/kg of bread are summarized in Table 2 and Table 3, respectively. Linearity for propionic acid was verified between 210 and 2519.7 mg/kg bread, and for sorbic acid between 195.7 and 2348.1 mg/kg bread. The limits of detection and quantification were 94 mg/kg and 314 mg/kg for propionic acid and 258 mg/kg and 849 mg/kg for sorbic acid. The mean recovery rate for propionic acid was 76.1%. For sorbic acid it was 57.5%. Table 2 shows that there was a loss of propionic acid during the analytical process, e.g., during the sample preparation and measurement steps. Most probably, the loss occurred during the distillation step, which is based on a rather old German Official Method developed in 1985. Table 3 similarly shows that admixtures 1 to 4 had very low recovery values (less than 70%). Hence, future research could work towards optimizing the sample preparation. On the other hand, the current procedure is quickly and easily to conduct and does not require further enrichment steps, because high dilution during distillation is avoided. The procedure rather under- than overestimates the content; therefore, we believe that exceedances of maximum limits would be provable with sufficient levels of certainty.

The use of propionic and sorbic acid reference standards enabled precise assignment of chemical shifts. The information used for identification and quantification of both acids is shown in Table 4 and Table 5. The characteristic signals can be assigned to the corresponding analytes unambiguously and without overlapping (Figure 1).

### 3.2. Propionic and Sorbic Acids Contents in Bread and Bakery Samples

A typical NMR spectrum of a positive sample including magnifications of target resonances is shown in Figure 2. Both compounds were observed in the low frequency region δ 1–1.3 ppm. The green spectrum represents an admixture with added standard substances while the red spectrum represents the sample.

From 76 fresh bread samples (i.e., not prepacked bread), none of the samples contained either sorbic or propionic acid. On the other hand, from 21 prepacked bakery wares, 10 samples contained sorbic acid, but none of them contained propionic acid (Table 6). It must be noted that the recorded sorbic acid concentrations, apart from in one sample, were lower than the maximum permitted levels (Table 6). A yufka sample contained both acids. The yufka sample was also exceptional, because it did not provide labelling of the contained acids, which is an infringement of the EU food information laws.

### 3.3. Measures Accompanying the Analysis

The method is suitable for clearly identifying propionic and sorbic acid in bread. The signals are specific. The method is also suitable for the quantitative determination of propionic acid in bread. It covers the usual concentrations for this matrix (undetectable up to the currently accepted maximum levels ranging between 1000 mg/kg and 3000 mg/kg). For sorbic acid, the method provides only guidance values due to low recoveries. The determined limits of quantification are sufficient for routine tests. Each measurement series was accompanied by quality control samples. The quality control charts (Figure 3) show sufficient long-term stability of the assay. The slight downward trend was judged as being due to potential loss of the acids due to evaporation or chemical degradation during storage rather than due to problems with measurement or spectrometers.

## 4. Discussion

Accompanying the technological development and modernization of food production and processing, the quality control of finished products has received the attention of a large range of scientific studies. These have been oriented towards the analysis of the effectiveness of the quality control as well as the quality and integrity of food additives. Despite this, fraud was constantly occurring in terms of incorrect labelling of the product or the type and origin of ingredients. Therefore, fully automated NMR spectroscopy, combined with multivariate data analysis techniques, has proven to be an efficient multipurpose tool to address these challenges [9,13,17].

As a matter of principle, recovery measurements are expressed in terms of the addition of a known amount of analyte to a sample and then determining the percent of the added amount. Food samples can therefore be spiked with varying amounts of a pure standard to give concentrations at the upper and lower reference limits [18]. Therefore, the recovery rate is usually given in percent and it describes the benchmarks for evaluating the quality of chemical analytical methods. It is defined as “the ratio of the amount of an analyte that is added to a sample before sample preparation and the amount of this analyte that is found as the measurement result” [16]. According to the same guidelines, if the recovery rate of an analyte is 100%, there has been no loss of this analyte during the entire analytical process (through particular analytical steps such as extraction and measurement). However, if the recovery rate is lower than 70%, this is an indication of substance loss. The percentage of losses can be verified by using additional internal standards or a recovery function [18,19].

Some evidence in the literature concluded that if the recovery observed for a spiked solution is identical to or iterates around 10% of the added concentration, the accuracy of the assay is considered as sufficient [19,20]. We have observed similar patterns in some spiked samples, especially in the case of propionic acid. As for sorbic acid, the recorded recovery rates did not fit the general IUPAC criteria [18].

In this study, the developed NMR method showed efficiency in detecting propionic and sorbic acids amounts in all analyzed bread samples. Nevertheless, we have observed that for both compounds, missing information were observed in the list of ingredients in some packaging (e.g., yufka samples). Furthermore, according to Article (18), paragraph (4) in connection with Annex VII, Part C of regulation (EU) No. 1169/2011 in regard to the provision of food information to consumers, food additives must be named according to their respective class. More attention should be drawn to the labelling and the use of the correct class designation in bread samples.

## 5. Conclusions

Our study is the first to provide NMR validation results for propionic and sorbic acid contents in bread and bakery products. The validated results showed that the developed method is quick and reproducible regarding the quantitative analysis of both substances. In order to use the developed NMR analysis for quality control of a broad range of cereals, further validation studies for other matrices (e.g., semolina, corn, doughs, dry mixes) is worthwhile. Nevertheless, further investigations of the sample preparation of both acids are necessary to increase the recovery rates, because the loss is expected to occur during the distillation step rather than the NMR measurement.

## Figures and Tables

**Figure 1 foods-10-00526-f001:**
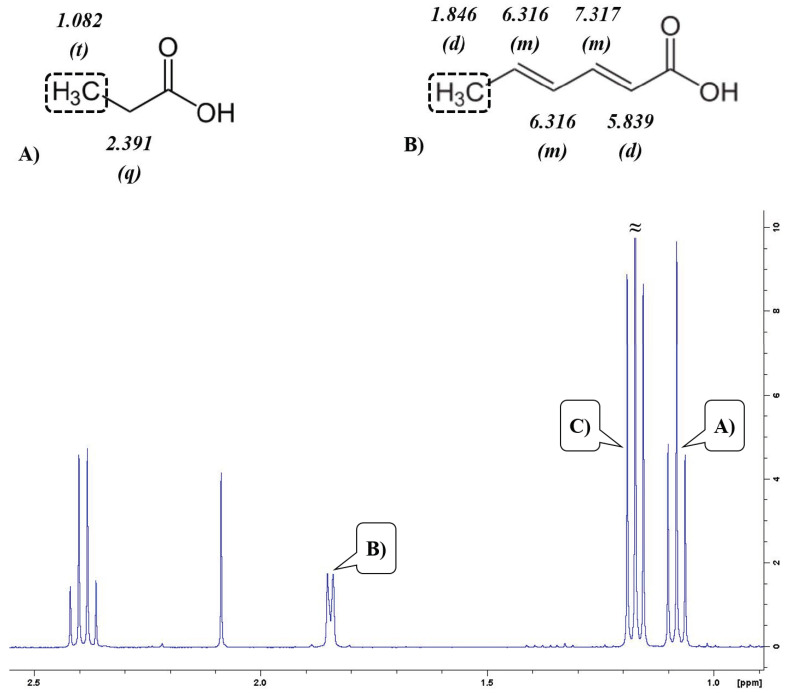
Structural formula of (**A**) propionic acid, (**B**) sorbic acid (upper panels) and assignment of ^1^H-NMR signals of A) propionic acid, B) sorbic acid, and C) ethanol–CH_3_, with top of triplet cut off (lower panel).

**Figure 2 foods-10-00526-f002:**
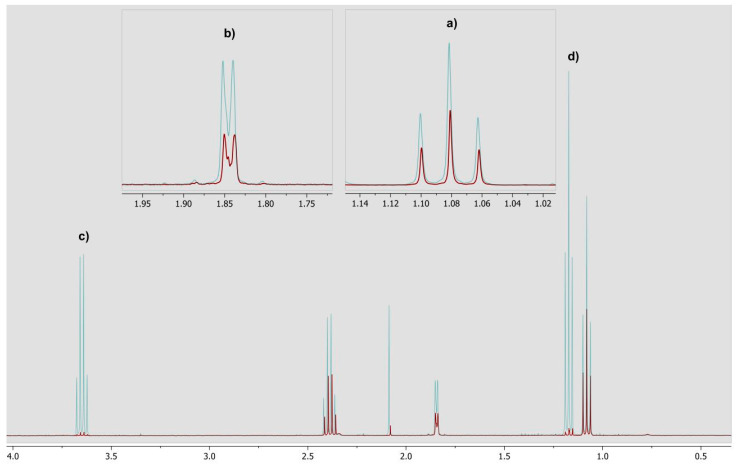
Representative ^1^H NMR spectra of target additives in yufka bread sample: (**a**) propionic acid; (**b**) sorbic acid (**c**) ethanol–CH_2_; (**d**) ethanol–CH_3_ (*x*-axis: ppm, *y*-axis arbitrary units).

**Figure 3 foods-10-00526-f003:**
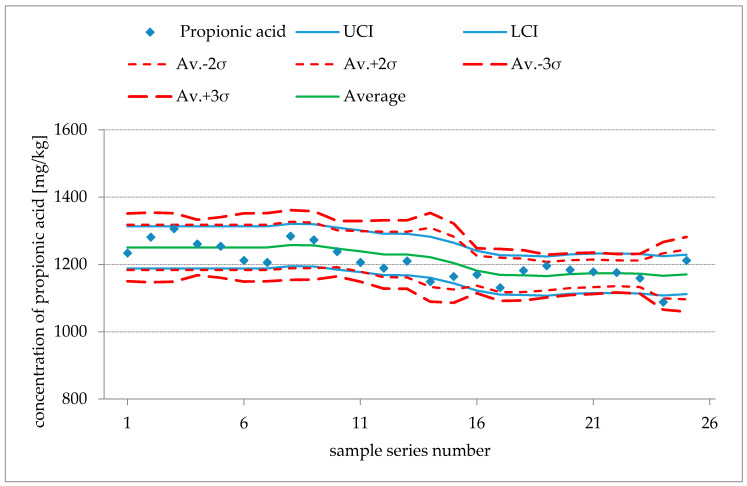
Quality control charts for propionic acid (upper panel) and sorbic acid (lower panel) (*x*-axis, sample series number; *y*-axis, concentration in mg/kg). By analyzing an aliquot of a reproducibility assessment sample with each new sample batch, for both analytes their respective rolling averages and accompanying spans of two and three standard deviations are calculated and monitored. Thus, deviations due to process faults can be detected. The range of 2·σ contains 95% and the range of 3·σ contains 99.7% of the probable results.

**Table 1 foods-10-00526-t001:** Maximum permitted levels (MPLs) of propionic acid/propionates (E 280–283) and sorbic acid/sorbates (E 200–202) in bread and bakery products according to the Annex II of Regulation (EC) No 1333/2008—levels are expressed as free acid [4].

	Bread and Bakery Type	Restrictions/Exceptions	Maximum Level (mg/L or mg/kg as Appropriate)
Propionic acid	Bread and rolls	Only prepacked sliced bread and rye bread	3000
	Bread and rolls	Only energy-reduced bread, partially baked prepacked bread and prepacked rolls, tortilla and pitta, prepacked polsebrod, boller and dansk flutes	2000
	Bread and rolls	Only prepacked bread	1000
	Fine bakery wares	Only prepacked fine bakery wares (including flour confectionery) with a water activity of more than 0.65	2000
Sorbic acid	Bread and rolls	Only prepacked sliced bread and rye-bread, partially baked, prepacked bakery wares intended for retail sale and energy-reduced bread intended for retail sale	2000
	Fine bakery wares	Only with a water activity of more than 0.65	2000

**Table 2 foods-10-00526-t002:** The theoretical and experimental concentration expressed in mg/kg as well as the recovery values (%) of propionic acid in bread samples.

Type	Concentration (in mg/kg)		Recovery (%)
	Experimental	Theoretical	
Bread without admixture	0	0	0
Admixture 1	205.6	260.4	79.0
Admixture 2	398	520.7	76.4
Admixture 3	777.8	1041.5	74.7
Admixture 4	2325.6	3124.4	74.4

**Table 3 foods-10-00526-t003:** The theoretical and experimental concentration expressed in mg/kg as well as the recovery values (%) of sorbic acid in bread samples.

Type	Concentration (in mg/kg)		Recovery (%)
	Experimental	Theoretical	
Bread without admixture	0	0	0
Admixture 1	157.2	242.6	64.8
Admixture 2	289.2	485.3	59.6
Admixture 3	486.2	970.5	50.1
Admixture 4	1618.2	2911.6	55.6

**Table 4 foods-10-00526-t004:** Chemical shifts, signal type, and number of protons for NMR identification of sorbic and propionic acid.

Substance	Propionic Acid		Sorbic Acid			
ppm	1.08	2.39	1.84	5.82	6.31	7.33
Signal type	Triplet	Quartet	Doublet	Doublet	Multiplate	
Number of protons	3	2	3	1	2	1

**Table 5 foods-10-00526-t005:** Information for NMR quantification of sorbic and propionic acid.

Analyte	CAS-No	M/(g/mol)	*δ*/ppm	Multiplet	J/Hz	Integration		*N*(H)
						from/ppm	to/ppm	
Propionic acid	79-09-4	74.08	1.082	t	7.5	1.117	1.045	3
Sorbic acid	110-44-1	112.13	1.846	d	5.0	1.875	1.820	3

**Table 6 foods-10-00526-t006:** Propionic acid and sorbic acid concentration expressed in mg/kg in all analyzed samples (n.d. not detectable (propionic acid: < 94 mg/kg; sorbic acid: < 258 mg/kg).

Concentration (mg/kg)				
Bread, Not Prepackaged (*n* = 76)	Yufka, Pasta (*n* = 2)		Rye-Bread, Partially Baked, Prepacked Bakery Wares (*n* = 21)	
	Sorbic Acid	Propionic Acid	Sorbic Acid	Propionic Acid
n.d. (all samples)	1183	1288	1005	n.d.
			1872	n.d.
			1453	n.d.
			1502	n.d.
			1770	n.d.
			2173	n.d.
			1399	n.d.
			1885	n.d.
			1235	n.d.
			1300	n.d.

## Data Availability

The data are not publicly available due to government policies.

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
