# Peer review of "Rapid Approach to Determine Propionic and Sorbic Acid Contents in Bread and Bakery Products Using 1H NMR Spectroscopy"

_foods, 2021, doi:10.3390/foods10030526_

Round 1
Reviewer 1 Report
This paper is a sound, concise application of NMR to in situ preservative quantification. There are some points that will need addressing;
Line 74 what were the conditions for storage and why was a 2 year gap chosen?
Line 135 What is the signal at 1.4ppm? Ethanol?
Line 167 Please explain the rationale behind using 2 and 3 standard deviations? What confidence interval are you aiming for? This should be better explained. A;sp the diagrams in figure 3 should have a similar vertical range for proper comparison of the the two analytes and the limits.
Line 170 Have you thought why the recovery for sorbic acid is low? Would either of these two preservatives be added during the baking process or subsequently?
You may wish to comment on potential decarboxylation or parasorbic acid formation? Has this been shown to occur during bread rising or baking? Was there any evidence for parasorbic acid in your spectra? This could be quite significant, given its relative toxicity.
Author Response
This paper is a sound, concise application of NMR to in situ preservative quantification.
Thank you!
There are some points that will need addressing;
Line 74 what were the conditions for storage and why was a 2 year gap chosen?
Information on sample storage was added. The two year gap can be explained by other obligations of our laboratory, so that we conducted two projects in 2017 and 2019.
Line 135 What is the signal at 1.4ppm? Ethanol?
Under given experimental conditions (esp. temperature and pH), ethanol shows resonances at 3.62 ppm (q) and 1.18 ppm (t), the triplet is visible in Fig. 1 and both resonances are visible in Fig. 2 (esp. the blueish spectrum). Ethanol is now indicated in the figures. The small multiplets between 1.45 ppm and 1.30 ppm could be from higher alcohols.
Line 167 Please explain the rationale behind using 2 and 3 standard deviations? What confidence interval are you aiming for? This should be better explained.
2SD and 3SD are widely used as diagnostic criteria in quality control charts, e.g. see: https://www.medcalc.org/manual/control_chart.php. As these limits contain 95% and 99,7% of the probable results, data inside these thresholds may be trusted as valid but outliers shall be analysed more deeply so as not to lose sight of a potential process fault. (Explanation added to fig.3).
A;sp the diagrams in figure 3 should have a similar vertical range for proper comparison of the the two analytes and the limits.
Due to the different concentrations of the two analytes used for reproducibility quality assessment, using the same full range for the ordinate would compress the graphs and encumber the visual evaluation. We matched the spans of the vertical ranges.
Line 170 Have you thought why the recovery for sorbic acid is low? Would either of these two preservatives be added during the baking process or subsequently?
The preservatives are typically incorporated into the dough mixture before baking. However, we believe that this does not influence on the recoveries because the baking step was not investigated. The discussion on recoveries was expanded.
You may wish to comment on potential decarboxylation or parasorbic acid formation? Has this been shown to occur during bread rising or baking? Was there any evidence for parasorbic acid in your spectra? This could be quite significant, given its relative toxicity.
We have not investigated parasorbic acid, which is the cyclic lactone of sorbic acid. According to literature, thermal treatment or hydrolysis converts the lactone to sorbic acid. Therefore, we expect that in baked products, the free form would be prevalent. As there still remains about 30-40% water in bread, re-cyclization is probably not expected.
Reviewer 2 Report
Dear Editor a send you my comments on the brief report entitled "Rapid approach to determine propionic and sorbic 3 acid contents in bread and bakery products using ¹H 4 NMR spectroscopy", Manuscript number foods-1116063.
Calcium propionate has been widely used as a preservative in bakery and in bread. It is sometimes not carefully used, or a high concentration is added to preserve products. High consumption of calcium propionate can lead to several health problems. The subject of this article is therefore of great interest.
For propionic acid, the quantitative determination was obtained by several cromatographic methods i. eGC-MS and HPLC .Although these methods have been accepted for their accurate and precise results, but they still have some drawbacks such as requiring expensive instruments and well-trained analysts. Moreover, toxic organic solvent waste was also produced
The report is well written, clear and it needs some minor revision. Ho
In particular the percentages of yield , reported in Table 2 and 3, change by increasing the distillation time which is about 4 minutes?
A clarification on this aspect could be added?
Author Response
Dear Editor a send you my comments on the brief report entitled "Rapid approach to determine propionic and sorbic 3 acid contents in bread and bakery products using ¹H 4 NMR spectroscopy", Manuscript number foods-1116063.
Calcium propionate has been widely used as a preservative in bakery and in bread. It is sometimes not carefully used, or a high concentration is added to preserve products. High consumption of calcium propionate can lead to several health problems. The subject of this article is therefore of great interest.
For propionic acid, the quantitative determination was obtained by several cromatographic methods i. eGC-MS and HPLC .Although these methods have been accepted for their accurate and precise results, but they still have some drawbacks such as requiring expensive instruments and well-trained analysts. Moreover, toxic organic solvent waste was also produced
The report is well written, clear and it needs some minor revision. Ho
Thank you!
In particular the percentages of yield , reported in Table 2 and 3, change by increasing the distillation time which is about 4 minutes?
A clarification on this aspect could be added?
We clarified this aspect. The problem is that increasing the distillation time also increases the dilution of the distillate or you need a larger receiver. This would then decrease the detection limit by dilution and also decrease the accuracy of the assay. There is already a rather large dilution (i.e. 10 g of sample to 100 mL of receiver volume). Going to larger volumes did not improve the samples preparation during some initial trials we have conducted.
Reviewer 3 Report
Dear Authors,
Bread is a basic and important component of our daily diet. Suitable food additives can be used to extend the shelf life. Sorbic acid and propionic acid may be safely used as antimicrobial agents. They are safe substances.
However, research to test the safety of their use in food products is very important.
The article submitted for review is succinct and correct. The presented studies provide the results of NMR validation for the content of propionic and sorbic acid in bread and bakery products. The advantage is that the developed method is fast and reproducible in terms of the quantitative analysis of both tested substances.
Author Response
Dear Authors,
Bread is a basic and important component of our daily diet. Suitable food additives can be used to extend the shelf life. Sorbic acid and propionic acid may be safely used as antimicrobial agents. They are safe substances.
However, research to test the safety of their use in food products is very important.
The article submitted for review is succinct and correct. The presented studies provide the results of NMR validation for the content of propionic and sorbic acid in bread and bakery products. The advantage is that the developed method is fast and reproducible in terms of the quantitative analysis of both tested substances.
Thank you!
Reviewer 4 Report
This is valuable research especially because it pointed out the false labeling of chemicals added to the breast paste and breast. A fast method for determination is always a good thing. I have no major remarks but to correct the units in Figure 3 (first graph) and to replace the decimal comma with a dot in several places (if necessary).
Author Response
This is valuable research especially because it pointed out the false labeling of chemicals added to the breast paste and breast. A fast method for determination is always a good thing.
Thank you!
I have no major remarks but to correct the units in Figure 3 (first graph) and to replace the decimal comma with a dot in several places (if necessary).
The unit was added in Figure 3 (first graph)! Thank you for spotting this mistake. The decimal commas/dots looks fine (English usage).